# Axonal Organelles as Molecular Platforms for Axon Growth and Regeneration after Injury

**DOI:** 10.3390/ijms22041798

**Published:** 2021-02-11

**Authors:** Veselina Petrova, Bart Nieuwenhuis, James W. Fawcett, Richard Eva

**Affiliations:** 1John van Geest Centre for Brain Repair, Department of Clinical Neurosciences, University of Cambridge, Cambridge CB2 0PY, UK; bn246@cam.ac.uk (B.N.); jf108@cam.ac.uk (J.W.F.); 2Cambridge Institute for Medical Research, University of Cambridge, Cambridge CB2 0XY, UK; 3Centre for Reconstructive Neuroscience, Institute of Experimental Medicine CAS, 142 20 Prague, Czech Republic

**Keywords:** axon growth, axon regeneration, inter-organelle membrane contact sites, organelles

## Abstract

Investigating the molecular mechanisms governing developmental axon growth has been a useful approach for identifying new strategies for boosting axon regeneration after injury, with the goal of treating debilitating conditions such as spinal cord injury and vision loss. The picture emerging is that various axonal organelles are important centers for organizing the molecular mechanisms and machinery required for growth cone development and axon extension, and these have recently been targeted to stimulate robust regeneration in the injured adult central nervous system (CNS). This review summarizes recent literature highlighting a central role for organelles such as recycling endosomes, the endoplasmic reticulum, mitochondria, lysosomes, autophagosomes and the proteasome in developmental axon growth, and describes how these organelles can be targeted to promote axon regeneration after injury to the adult CNS. This review also examines the connections between these organelles in developing and regenerating axons, and finally discusses the molecular mechanisms within the axon that are required for successful axon growth.

## 1. Introduction

Neurons of the peripheral and central nervous systems (PNS and CNS) represent a population of highly polarized cells, with axonal terminals usually located long distances away from the cell bodies. This unique morphology is crucial for proper neuronal function but also calls for intricate regulation of the processes governing axon growth and regeneration after injury.

During early development, neuronal processes undergo rapid growth with one process being specified as the axon [1,2,3,4]. The axon grows from its tip, the growth cone, and extends long distances through the extracellular environment. The growth cone is a highly dynamic structure which guides axons to their correct target by sensing, integrating and responding to signals from chemoattractive and chemorepellent cues [5]. In some human neurons, the axon could extend up to 1m in length. The intricate process of axon growth requires enormous amounts of lipid synthesis, endosomal trafficking and correct positioning of materials and machinery at the plasma membrane in order to drive the axon forwards [6]. Furthermore, tight regulation of cytoskeletal dynamics and integrated intracellular signaling are key for successful axon growth, target identification and synapse formation [7].

Unlike PNS and embryonic CNS neurons, adult CNS neurons have very limited regenerative capacity and, therefore, re-grow poorly after injury. The mechanisms required for axonal extension after injury in many aspects mimic the processes that occur during developmental axon growth; however, the regenerative growth process is further complicated by the primary requirement for growth cone redevelopment. Recent studies have shown that stimulating developmental growth programs in the adult is one way of restoring axonal growth and function ([8] and summarized in [9,10]); however, there are a number of issues that need to be addressed in adult CNS neurons. For example, as neurons mature and form synapses, synaptic maintenance trumps growth and, as a consequence, the cell adjusts the availability and localization of the axon growth machinery to fit these newly found needs (reviewed in [11]). In addition, axonal injury can occur at long distances away from the cell body and, therefore, successful regeneration requires active axonal transport as well as local regulation of protein and lipids [12]. The extracellular environment after injury also differs for mature CNS neurons compared to the time of developmental CNS growth and to injured PNS neurons, with numerous inhibitory factors impeding growth after injury [13].

One requirement for the initiation of successful regrowth programs in the PNS after injury is the resealing of the plasma membrane and the formation of a new growth cone. These processes are highly dependent on calcium influx [14,15]. Calcium affects numerous intracellular signaling processes which communicate to the cell body to mount a regenerative response through the expression of regeneration-associated genes (RAGs), cytoskeletal remodeling or axonal transport [16,17]. A number of key intracellular signaling pathways have so far been identified that contribute to the regulation of axon regeneration including the PTEN/PI3K, the JAK/STAT, the ERK/MAPK and the RhoA/ROCK pathway [18,19,20,21,22,23]. However, targeting individual signaling pathways is insufficient to elicit optimal regeneration in the adult CNS and highlights the need for further approaches to enable the robust, long-range regeneration needed for functional recovery after injury.

Recent advances in microscopy and live-imaging techniques have highlighted that intracellular organelles are important signaling platforms and sources of the materials required for axon growth and regeneration. This review summarizes the evidence for intracellular organelles as sites of convergence for many growth and regeneration-associated pathways, discusses the numerous potential roles that inter-organelle connections might play during axon growth and regeneration, and debates how individual organelles or multi-organellar complexes could be targeted to stimulate axon growth. We review the current literature on the involvement of the endoplasmic reticulum (ER), mitochondria, the endo-lysosomal system and the proteasome during axon growth and regeneration. We also summarize the intricate interactions between different organelles and their significance in axon health and disease.

## 2. Endoplasmic Reticulum

### 2.1. Structure and Function

The endoplasmic reticulum (ER) is a membranous organelle that spans across the entire intracellular space and has essential physiological functions such as the synthesis and recycling of lipids, the maintenance of calcium homeostasis, metabolite processing as well as the synthesis and redistribution of secretory or membrane-associated proteins. The ER is composed of two distinct interconnected structures—ER tubules and flattened ER sheets. The latter could, in some cases represent a densely packaged network of tubules [24,25,26]. This structure supports a highly dynamic model where the ER can rapidly change its structure and distribution to meet any changing cellular demands and to modulate its interactions with various other organelles throughout the cell, making it a perfect candidate for a role in axon growth and regeneration.

In the cell body, most ER is studded with ribosomes (referred to as rough ER) which reflects its essential role in protein translation. The ER throughout the axon constitutes a continuous ribosome-free membranous structure of smooth ER tubules, which run in parallel to the axonal plasma membrane [27,28]. In some cases, an individual tubule has been found to span across the entire length of very thin axonal processes while at the same time maintaining its connection with ER in the rest of the cell [29,30,31]. This interconnectedness between differently shaped ER structures is essential for axon integrity and function [32] and is achieved by the function of ER-shaping proteins [33,34]. Mutations in ER-shaping proteins result in human neurological disease, highlighting their importance for proper axonal function [35]. For example, several mutations in hairpin-loop containing ER-shaping proteins such as reticulons, REEPs and atlastins have been described as causative of autosomal dominant hereditary spastic paraplegia (HSP), a condition where distal axons progressively degenerate resulting in severe motor and sensory symptoms [36,37]. Recent studies utilizing *Drosophila* models have shown that mutations in the reticulon and REEP families of ER-shaping proteins do indeed result in ER fragmentation and network disruption with physiological consequences in the distal axon, demonstrating that ER shaping and remodeling have direct effects on axon structure, function and maintenance [38,39,40].

The studies above indicate that a continuous, intact ER is required throughout the axon, and that disruption of axonal ER leads to degeneration. ER integrity could also have important implications in axon regeneration—rebuilding the structure and function of axonal ER after injury might be just as important as rebuilding the structure of the axon itself, perhaps even being an essential prerequisite before growth cone redevelopment. Examining the function and anatomy of the ER during developmental axon growth can help to understand how it might participate in successful axon regeneration.

### 2.2. ER Shaping and Distribution in Axon Growth and Regeneration

Many interactions are involved in shaping the ER and determining its distribution. Studies manipulating ER-related proteins have highlighted that the ER is an important organelle for developmental axon growth. In neurons, the tubular shape of the ER along the axon has been suggested to be favorable for the trafficking and function of other organelles and cellular components during rapid growth, therefore ER-shaping proteins are also desirable in those axon compartments [41]. ER-shaping proteins have indeed been found to be enriched throughout the axon and in the growth cone of developing [42,43,44,45] and regenerating axons [46,47]. For example, reticulons and atlastins are distributed throughout axons and dendrites of developing hippocampal neurons in culture [42] and at the growth cone of cultured cortical neurons where their depletion reduces axon growth in early development [48,49]. In addition, REEP1 colocalizes with atlastin and spastin in the growth cone, axonal varicosities and in the axon shaft of cortical neurons in culture [44]. In fact, knockdown of REEP1 or overexpression of an HSP-associated mutant form of REEP1 leads to impaired neurite outgrowth and results in axon degeneration in mouse primary cortical neurons [45]. Another study found enrichment of REEP5 and reticulon 4 proteins at the axonal growth cone of neurons overexpressing an active version of an ER-associated adaptor protein called protrudin, and this was associated with increased neurite outgrowth [43]. In addition, the reduction of protrudin’s interaction with ER-associated proteins such as VAP-A has been shown to reduce axon growth in culture [50]. The ER has also been found to accumulate at both the proximal and distal part of newly growing neuronal-like processes in PC12 cells during a rapid state of growth, initiated by a directed dragging force with the use of nanoparticles [51]. The distribution and structural organization of ER tubules along the axon was also shown to be closely linked to microtubule stability and vice versa; this interaction was found to be critical for the establishment of neuronal polarity and axon specification during early development of hippocampal neurons [42].

A recent study in *Drosophila* suggests that ER, cytoskeletal dynamics and localization at the growth cone could be essential requirements for successful axon regeneration. In this study, ER and spastin proteins accumulated at the tips of regenerating axons after axonal but not dendritic injury [46]. Mutant flies defective for atlastin or spastin did not show ER enrichment at the growth cone after injury, which was concordant with impaired axon regeneration [46]. Another study found that exogenous reticulon-1-GFP, a fluorescent marker of smooth ER, localized to the tips of regenerating axons, but again not dendrites, in a model of a simultaneous injury to all processes of mature sensory neurons in *Drosophila* [47]. Interestingly, both axons and dendrites re-grew after injury and no growth deficits were observed in comparison to each process regenerating alone. In mammalian studies of axon regeneration, overexpression of the ER resident and shaping protein protrudin resulted in increased amounts of ER tubules in the distal axon and at the growth cone, and this was accompanied by extensive axon regeneration in rat cortical neurons after laser axotomy in vitro and in mouse retinal ganglion cells (RGCs) after optic nerve crush in vivo [43]. The regenerative effect of protrudin overexpression was abolished when its localization within the ER or its interaction with the ER contact site proteins VAP-A/B were removed [43].

### 2.3. ER in Lipid Synthesis and Trafficking during Axon Growth and Regeneration

Lipids are the major building blocks of all membranes. In some developing neurons, the plasma membrane surface area can expand up to 20% per day, a process which requires tightly regulated mechanisms for lipid synthesis and trafficking [6]. The ER is a major site for axonal biosynthesis of lipids such as phosphatidylcholine, sphingomyelin, phosphatidylethanolamine as well as fatty acids and phosphoinositides [52,53]. In fact, nearly half of the axonal phosphatidylcholine is locally synthetized in the distal axon and this synthesis is required for axon growth [54,55]. In addition, most major lipid synthetizing enzymes reside within the ER and their activity is regulated by feedback signaling from target organelles to the ER which is important for proper growth during development [56,57]. The sterol regulatory element-biding proteins which are resident ER-proteins can feedback on their own synthesis and this is a common mechanism through which pro-growth mTOR signaling can influence the production of lipids in the ER [58,59,60]. There are two major pathways through which lipids are transported from the ER to the plasma membrane (PM) during rapid axonal growth—vesicle-dependent and vesicle-independent transport. Vesicular transport of lipids is discussed in more detail in the section below on protein synthesis and trafficking.

Non-vesicular transfer of lipids from the ER to the PM depends on lipid transfer proteins acting at contact sites between the ER and the PM [61,62,63]. This mode of lipid delivery allows for fast and efficient insertion of locally synthetized lipids into rapidly expanding membranes during axonal outgrowth in comparison to vesicle transport [64]. Recent studies suggest that one mechanism of bulk lipid transfer during axon growth is driven by a protein complex consisting of at least ER-localized SNARE protein Sec22b and a plasma membrane SNARE protein syntaxin1 (Stx1) at the growth cone to create a non-fusogenic bridge between the ER and the PM [65,66]. This lipid transfer and protein complex was recently shown to be regulated by extended synaptotagmin (E-Syt) which is an ER-resident lipid transfer protein. Overexpression of E-Syt in developing neurons dramatically enhances neurite outgrowth and neurite ramification by stabilizing the Sec22b-Stx1 interaction and providing contact sites for lipid insertion to drive new membrane expansion [67]. Importantly, this process depends on E-Syt being localized within the ER [67].

### 2.4. ER in Protein Synthesis and Trafficking during Axon Growth and Regeneration

A historical view of axon growth suggests that signaling proteins, growth receptors and other growth-promoting molecules are exclusively synthetized in the cell body in the rough ER and transported to the developing or regenerating axon via microtubule-based transport [68]. Indeed, whilst the majority of membrane proteins are most likely transported this way, recent studies have identified that proteins can also be synthetized locally, allowing for more efficient transport to the growth cone or the site of injury in response to the metabolic demands of the cell changing rapidly upon targeted axon growth or injury [30,69,70,71,72,73,74,75,76]. Despite the lack of conventional Golgi structures in neuronal processes, numerous elements of the protein synthesis pathway have been detected throughout axons and dendrites. Those include mRNAs, scattered ribosomes, polysomes and translational chaperones [77]. It is possible that many of these elements as well as other translational platforms such as endosomes and mitochondria associate with the ER in order to regulate this process [30,72,75]. Further studies are, however, needed in order to determine the exact role of axonal ER in local protein translation, particularly as axonal ER appears smooth and tubular, unlike the ribosome studded rough ER found in cell bodies. One way in which the ER has been shown to influence local axonal translation is through its regulation of calcium dynamics. For example, axonal injury can trigger calcium release from ER stores, which in turn could stimulate translation of axonal proteins aiding the regenerative response in PNS neurons [78,79].

The trafficking and export of newly synthesized lipids and proteins in vesicular structures from the ER to the cell surface is a key mechanism that can influence axon growth and development [55,80]. ER-resident proteins can influence this process. For example, downregulation of atlastin in motor neurons in *Drosophila* results in altered secretory pathways and presynaptic protein distribution as well synaptic vesicle release, which ultimately resulted in behavioral deficits [81]. In addition, overexpression of reticulon-2 increased the delivery of glutamate transporter (EAAC1) from the endoplasmic reticulum to the plasma membrane [82]. Furthermore, VAP-A can be internally cleaved in the ER to initiate interaction with growth cone guidance receptors such as ephrins, Robo and Lar in order to influence axon guidance and pathfinding in the developing brain [83]. Interestingly, VAP-A was also shown to interact with tethering proteins such as Sec22 and Stx1 (which are mentioned earlier with regard to the non-vesicular trafficking of lipids) to stabilize PM-ER contact sites [67]. Another ER-resident protein, IER3IP1 was recently shown to regulate the ER ubiquitin-proteasome system (UPS) response and the secretory extracellular matrix molecules, with mutations leading to microcephaly and impaired developmental growth [84]. The trafficking and distribution to the cell surface of signaling receptors and ion channels such as NMDA [85], GABA [86] and AMPA receptors [87] were all shown to be ER-dependent.

### 2.5. ER Calcium Buffering during Axon Growth and Regeneration

Another major role for the ER during axon growth and regeneration is calcium buffering and regulated calcium release. The targeted elongation of axons during neuronal development in response to attraction cues is coupled to calcium release from the endoplasmic reticulum, which, in turn, aids the transport and exocytosis of VAMP2-positive vesicles on the side of the growth cone that has elevated calcium [88,89]. Another study demonstrated that Myosin Va can act as a sensor of ER-derived calcium to drive the release of membrane vesicles, which it normally tethers to two major ER channels—the ryanodine receptor type 3 (RyR3) and the 1,4,5-triphosphate (IP3) receptor (IP3R) [90]. This process was shown to be instrumental for attractive growth cone turning and for proper development of the chick spinal cord.

ER calcium buffering is an important step to allow for developmental growth and axon regeneration. For example, ER calcium-sensor STIM1 (stromal interaction molecule 1), localized with MT-plus-end binding proteins EB1/EB3 in growth cones upon BDNF exposure in developing zebrafish motor neurons [91]. STIM1 in turn promoted active ER remodeling, calcium signaling and growth cone steering during developmental axon guidance [91]. In addition, STIM1 and E-syts are actively involved in the formation of ER-PM contact sites [67], replenishment of ER calcium and the clustering of diverse proteins at the PM which has major implications for growth and regeneration [92,93]. In injury, ER-regulated backpropagating calcium wave is required for the regenerative response in DRG neurons [94]. In addition, peripheral nerve injury in mouse DRG spot cultures accelerated ER calcium release, activated the unfolded protein response (UPR) at the injury site and aided new growth cones formation [95]. UPR activation after a sciatic nerve crush also promoted axon regeneration in DRG axons [96]. The role of ER-resident stress transducers such as IRE1 (Inositol-requiring enzyme 1) and CHOP (C/EBP-homologous protein) have also been implicated in motor neuron response after spinal cord injury [97] and in RGCs response after optic nerve injury [98]. The role of ER stress and the UPR response after injury and neurodegenerative disease have been extensively reviewed elsewhere [99,100].

In addition to ER’s involvement in the regulation of growth, the ER could also act as a molecular platform for neurotransmission dynamics and proper brain function. Axonal ER calcium content as detected by STIM1, was previously shown to control neurotransmitter release at CNS axon terminals [101]. In addition, VAP-A interactions with secernin-1 (SCRN1) at the ER membrane was recently shown to not only be important for ER continuity but also for proper synaptic vesicle release and calcium homeostasis [102]. Furthermore, disruption in axonal autophagy was shown to result in increased tubular ER in axons leading to the dysregulation of calcium stores and aberrant synaptic activity [103].

Taken together, the endoplasmic reticulum appears to be a dynamic structure supporting numerous processes important for axon growth and regeneration such as lipid and protein synthesis as well as calcium homeostasis while at the same time acting as a platform for a variety of proteins important for signaling and response to injury.

## 3. Mitochondria

### 3.1. Structure and Function

Mitochondria are double-membrane organelles which were termed “the powerhouse of the cell” due to their role in ATP (adenosine triphosphate) production. However, mitochondria are also important for calcium buffering, redox homeostasis, apoptosis and cell death signaling [104,105,106]. The proper localization and functioning of mitochondria are important to meet the high energy demands of developing neurons and to support neuronal processes such as axon growth and neurotransmission. Recent advances in imaging have allowed for the visualization of mitochondria in vitro and in intact organisms such as mice and zebrafish [26,31,107,108,109,110,111]. Similar to other organelles, mitochondrial transport into the axon is mostly microtubule-based and relies on motor proteins such as kinesin and dynein as well as mitochondria-specific adaptor proteins such as Milton and Miro [112,113,114,115,116]. Importantly, these live-imaging studies showed that axonal transport of mitochondria is highly dynamic but also developmentally down-regulated in the CNS. In developing neurons, mitochondria are highly mobile, move bidirectionally and preferentially localize to the active growth cone [116,117,118]. In mature neurons, mitochondria move more slowly, exhibit elongated morphology and the majority of mitochondria are anchored in stationary positions along the axon [107,110,118,119,120]. In addition, mitochondria can be regulated by proteins such as PTEN, BDNF, mTOR, NGF, NOGO, semaphorins and CSPGs, which are all well-known molecules involved in signaling during neurite growth, axon regeneration and maintenance of synapses [121,122].

### 3.2. Mitochondrial Fission and Fusion in Axon Growth and Regeneration

Regulation of mitochondrial size by the processes of mitochondrial fusion and fission is an important determinant of developmental axon growth. Deletion of key fusion and fission proteins, such as MFN1, MFN2 and Opa1, leads to developmental deficits such as impaired axon growth and is lethal [123,124]. Increased mitochondrial fusion and decreased fission by DRP-1 (dynamin-like protein) downregulation was shown to suppress neurite outgrowth in RGCs, to decrease growth rate and to prevent growth cone turning on inhibitory substrates [121]. In the same study, mitochondrial size dynamics were shown to be developmentally down-regulated in RGCs, an event which coincides with the reduction in intrinsic axon growth and regeneration capacity [121]. Similarly, downregulation of mitochondrial fission factor (MFF) results in elongated mitochondria with altered calcium buffering capacity and this decreased neurotransmission and reduced terminal axon branching in cortical pyramidal neurons [125]. This study pinpoints the importance of maintaining small axonal mitochondria during neurotransmitter release and axon development. In addition, shorter and more mobile mitochondria were observed in regenerating motor axons after nerve injury and in retinal neurons after an optic nerve crush suggesting the importance of early mitochondrial fission after axonal injury [126]. Indeed, ablating mitochondrial fission resulted in elongated mitochondria, which also impaired mitochondrial functions and caused neuronal cell death [126]. Contradictory to the findings above, knockdown of mitochondrial pro-fission protein MTP18 increased mitochondrial size but also promoted axonal outgrowth in RGCs cultured on inhibitory substrates [122]. This approach had however no effect on axon regeneration or RGC survival after optic nerve crush in vivo [122]. Taken together, these results indicate that the optimal balance between mitochondrial fusion and fission might be cell type-specific, injury-dependent and cell compartment-driven but nonetheless, underline the importance of mitochondrial size and in its involvement in axon growth.

Mitochondrial biogenesis, energy production and transport into the axon are crucial mechanisms supporting axon growth. A hallmark study found that depletion of mitochondria prevented axon specification and polarization [127]. In cultured cortical neurons, overexpression of PGC-1α, a regulator of mitochondrial biogenesis and energy production, resulted in elongated axon and dendrites, while inhibition of glycolysis reduced oxidative phosphorylation or knockdown of PGC-1α all inhibited neurite outgrowth [128]. Another study found that overexpression of PGC-1 increased mitochondrial density, improved its redox state and delayed Wallerian degeneration in a neurodegenerative model of zebrafish [111]. Axonal mitochondrial transport has a key role in the process of developmental axon growth. Deletion of motor adaptor Trak1 resulted in inhibited axon outgrowth and branching in cultured hippocampal neurons [129]. Deacetylation of adaptor protein Miro1 by HDAC6, a histone deacetylase, decreased mitochondrial transport and this was in line with reduced axonal growth of adult DRG neurons on growth-repulsive substrates [130].

### 3.3. Mitochondrial Transport in Axon Growth and Regeneration

In addition to their participation in developmental processes, mitochondria have been implicated as key players in axon regeneration and degeneration. Axonal injury results in mitochondrial depolarization and oxidative stress, processes which are associated with neurodegeneration and cell death [111,131]. Therefore, there is a need for functioning mitochondrial transport not only to remove unhealthy mitochondria but also to supply functioning ones for repair [118,132]. After axonal injury, key regenerative processes such as for membrane resealing, growth cone formation and axon extension require energy. Due to the limited capacity of intracellular ATP for long-range transport across the cell, efficient mitochondrial trafficking and the local production of ATP near the site of injury are key to successful regeneration [120]. Previous studies have shown that there is increased trafficking of mitochondria in injured but regenerating axons in vivo after peripheral nerve injury [109,110] whereas dramatically reduced anterograde flux of mitochondria after injury was associated with poor regeneration and degeneration of the distal axon [110]. There is a possibility that the developmental decline in axonal transport of mitochondria in mature cells of the CNS could partly explain the regenerative failure seen in many adult models of axonal injury.

Studies into mitochondrial positioning and axon regeneration have identified a number of key regulatory molecules, including Armcx1 and syntaphilin. Armcx1 was identified after comparative gene expression profiling of non-regenerative and highly regenerative RGCs [133]. Overexpression of Armcx1, a mitochondria resident protein, resulted in increased mobility of mitochondria, which in turn promoted neurite outgrowth and survival not only in cultured embryonic cortical neurons but also in adult RGCs after optic nerve crush [133]. Interestingly, when Armcx1 is knocked out in a highly regenerative genetic model, both axon regeneration and cell survival are inhibited [133]. In a different study, mitochondria were found to be less mobile with maturation which was attributed to increased expression of mitochondrial anchor protein—syntaphilin and correlated with reduced regenerative capacity [118]. Knockout of syntaphilin in DRG neurons resulted in increased influx of mitochondria into the axon and increased axon regeneration both in vitro after axotomy and in vivo after sciatic nerve crush [118]. Interestingly, syntaphilin knockout did not result in developmental changes in axon growth. Syntaphilin knockouts have been used since to show increased corticospinal tract axon regeneration through a spinal cord lesion, improved regrowth of monoaminergic axons, increased sprouting and functional recovery in three different models of CNS injury [134]. Administering creatine—an energy facilitator—further exaggerated this effect, suggesting that improving mitochondrial transport and restoring cellular energetics is sufficient to promote regrowth and functional recovery in the injured CNS.

Mitochondrial transport as a regulator of regeneration has also been studied in *C. elegans*. In vivo laser axotomy of single axons showed a two-fold increase in the amount of mitochondria in the axon after injury where axons with low mitochondria density did not regenerate and axons with high mitochondria density showed robust regenerative response [112]. The regenerative response was dependent on mitochondrial transport in the axon as its blockade by Miro-1 RNAi prevented mitochondrial accumulation and regeneration whereas overexpression of Miro resulted in mitochondrial delivery to the axon and improved regeneration [112]. This injury-induced response was regulated by activation of DLK-1 (dual leucine zipper kinase 1)—a conserved axon regeneration kinase. Additionally, *C. elegans* in which the respiratory chain was defective also failed to regenerate implicating the importance of energy production for growth [112]. Indeed, elements of the electron transport chain were found to be essential for the regrowth of mechanosensory axons in *C. elegans* after laser axotomy; although, in this paradigm, mitochondria did not increase at the growing tip and defective fusion/fission genes did not alter regeneration [135]. In addition, *C. elegans* mutants lacking *ric-7*—a gene essential for mitochondrial localization—failed to transport mitochondria into the distal axon, which resulted in rapid axon degeneration after injury [136]. This effect could be completely reversed when mitochondria are supplied to the axon.

### 3.4. Mitochondrial Calcium Dynamics in Axon Growth and Regeneration

As discussed above in relation to the ER, transient influx of calcium after injury is an important factor for axon regeneration. Axonal mitochondrial calcium levels are regulated by MCU-1—a mitochondrial calcium transporter. In *C. elegans,* a deletion of translational repressor—CAR1 resulted in increased levels of MCU-1 which, in turn, resulted in a more sustained influx of calcium in axonal mitochondria and enhanced axon growth after injury [137]. In addition, both Miro and syntaphilin possess calcium-binding domains, which could result in regulation of axonal transport of mitochondria by surrounding calcium levels [138,139]. These studies underline the importance of mitochondrial calcium dynamics after injury for the processes of growth and regeneration.

### 3.5. Mitochondria as Molecular Platforms for Axon Growth and Regeneration

Additionally, mitochondria can also act as molecular platforms where various proteins can anchor and signal to influence the processes of growth and regeneration. Several kinases have been implicated in regulating mitochondrial function and dynamics. For example, the LKB1-NUAK1 pathway is involved in mitochondrial anchoring and immobilization in axons, which is a process essential for terminal axonal branching [140]. Activation of AMPK—a master regulator of cellular dynamics and ATP production during growth results in increased influx of mitochondria into the axon and induces axon branching in regions of high mitochondrial availability [141]. STAT3 is a transcription factor previously described as a promoter of axon growth and regeneration in the CNS [142]. In addition to its transcriptional activity, its localization within mitochondria was recently shown to improve their ATP production properties and support axon growth after optic nerve crush and spinal cord injury [143]. This effect was regulated by MEK kinase and further potentiated by PTEN deletion. NMNAT, an NAD+ synthetizing enzyme is another molecule functioning within mitochondria that support proper axonal function. For example, upregulation of NMNAT suppresses degeneration in adult *Drosophila* by alleviating injury-induced mitochondrial loss [144]. In fact, targeting NMNAT to mitochondria is fully sufficient to mimic the effects of Wlds mutation which was found to drastically delay Wallerian degeneration in both mice [145,146] and *Drosophila* models after injury [146]. These effects were attributed to improved calcium buffering properties of mitochondria after injury and increased mitochondrial motility into axons.

In summary, mitochondria are organelles with diverse cellular functions that are essential for developmental axon growth and regeneration after injury and that could act as molecular platforms for intracellular signaling governing these processes.

## 4. Endosomes

### 4.1. Structure and Function

The endocytic pathway regulates the internalization of membrane and membrane-associated proteins into endosomes and their trafficking to and from the plasma membrane. These components are either recycled, transported to their target location within the cell, or sorted towards degeneration via late endosomes and the lysosome. Hence, endosomes can be targeted to different parts of the cell and the endocytic pathway can, therefore, exert a strong control over the localization of specific cellular machinery. The endocytic pathway also influences axon growth and regeneration by controlling cellular events such as intracellular signaling, membrane recycling, protein sorting and degradation [12,63,147,148,149,150]. The transport and function of endosomes is controlled by small GTPases such as the Rab, ARF and ARL families [151], which cycle between GDP- and GTP-bound states as a result of being regulated by a variety of activating or inhibiting factors. Specific GTPases mark and regulate specific endosomal populations. For example, early endosomes can be marked by Rab5, late endosomes by Rab7 and recycling endosomes by Rab4 or Rab11 [152,153,154]. Here, we summarize the role of key Rabs in axon growth and regeneration and some of the mechanisms through which they can govern these processes.

### 4.2. Endosomal Regulation by Rab11 in Axon Growth and Regeneration

Perhaps the most well-studied Rab during axon growth and regeneration is Rab11. Optogenetic targeting of different cytoskeletal motors to Rab11-positive recycling endosomes revealed that the addition of dynein motors resulted in removal of endosomes from the growth cone and reduced axon growth. In contrast, loading of kinesin onto these endosomes stimulated their transport towards growth cones and this dramatically improved axon growth in primary hippocampal neurons [155]. Overexpression of Rab11 or its effector—Rab-coupling protein—resulted in increased anterograde trafficking of growth-related integrin receptors to the cell-surface of adult DRG neurons [156] and in differentiated PC12 cells [157]. Rab11-positive endosomes colocalize with TC10, another GTPase involved in endosome recycling and vesicular fusion, in growth cones and undergo plasma membrane insertion via the exocyst complex [158]. The anterograde transport of Rab11-positive endosomes and the plasma membrane insertion potentiated axon growth in vitro [156,157,158]. Rab11-dependent endosomal transport was shown to be inhibited by the kinase LMTK1 in mouse cortical neurons where downregulation of LMTK1 had dramatic effects in aiding Rab11 trafficking to the growth cone and stimulated both axonal and dendritic growth and branching [159,160,161]. Rab11-positive endosome are preferentially localized in the somatodendritic compartment of mature CNS neurons in culture [162], and overexpression of Rab11 in vitro leads to increased anterograde transport of recycling endosomes in the axon and improved regeneration after laser axotomy in vitro [162].

Rab11 regulates recycling endosomes through a close interaction with one member of the ARF family of small GTPases, ARF6. Rab11 and ARF6 form a complex with the transport adaptor JIP3/4, the activation state of ARF6 determining whether the complex will associate with kinesin for anterograde transport or dynein for retrograde travel [163]. ARF6 has also been identified as a regulator or axon regeneration. Inactivation of ARF6 resulted in increased integrin-containing endosomal trafficking to the plasma membrane and aided integrin-mediated neurite outgrowth in DRG neurons [164]. In matured CNS neurons, integrin and Rab11 trafficking into the axon is prevented by EFA6, an ARF6 activator which localizes to the axon initial segment [165]. Knockdown of EFA6 in cortical neurons permits the transport of Rab11 endosomes containing growth receptors to the growth cone and enhances not only initial axon growth but axon regeneration as well [166].

Rab11 has also been targeted to enhance regeneration in vivo. Viral delivery of either protrudin (an ER adaptor protein) or p110 delta (a natively hyper-active catalytic subunit of phosphoinositide 3-kinase) enhanced regeneration in the optic nerve, and were accompanied by enhanced transport of Rab11 into the distal part of the axon [20,43]. Protrudin has previously been shown to assist the movement Rab11-positive endosomes to the tips of PC12 cell neurites and can also stimulate the formation of cellular protrusions in non-neuronal cells [157]. It was recently demonstrated that overexpression of wildtype or activated protrudin leads to increased axonal transport of Rab11 in cortical neurons in vitro, as well as robust, long-range regeneration after an optic nerve crush injury [43]. In addition, overexpression of p110 delta in cortical neurons leads to increased anterograde transport of Rab11, partly through a down-regulation of ARF6 activity, but also by signaling via mTOR and CRMP2. This approach enabled regeneration after an optic nerve crush injury, identifying p110 delta overexpression as an alternative approach to boosting axon regeneration through the PI3K/PTEN pathway and manipulation of regenerative transport of Rab11-positive endosomes [20].

### 4.3. Endosome Regulation by Other Rabs in Axon Growth and Regeneration

A number of other Rab family GTPases have been shown to influence endosome function in plasma membrane expansion and receptor insertion during developmental axon growth [150,167,168,169]. Rab5 and Rab4 were both shown to co-localize with endosomes at *Xenopus* RGC growth cones where they are locally recruited to assist membrane recycling [170]. Mutations that disrupt Rab4 or Rab5 function resulted in decreased axon growth both in vitro and in vivo, but did not alter pathfinding, indicating a role for early endosomes in developmental axon extension [170]. These results were recently confirmed by another study which found that expression of mutant forms of Rab5 impaired axon growth in cultured *Xenopus* RGCs [30]. Mutations in Rab7, which are also associated with human neuropathy, were shown to cause developmental defects in axon growth, branching and pathfinding of sensory neurons in zebrafish [171]. Rab33a colocalizes with and regulates the transport of synaptophysin-positive vesicles destined for membrane insertion during neurite outgrowth [172]. Downregulation of Rab33a resulted in a decreased number of vesicles at the growth cone and reduced axon growth in rat hippocampal neurons [172]. Rab35, another GTPase involved with cargo loading onto recycling endosomes was also shown to play an important role in axon outgrowth in rat primary neurons [173]. Rab35 is degraded by a p53-related protein kinase (PRPK) through the UPS which negatively impacts axon growth. This degradative process is regulated by MAP1B—a microtubule-associated protein which blocks PRPK action and allows for Rab35-assisted axon extension [173]. This study highlighted the interplay between endosomal trafficking, the cytoskeleton and protein degradation in the process of axon growth. Furthermore, the anterograde transport and vesicle fusion of post-Golgi Rab10-positive vesicles carrying various growth receptors were shown to be essential for neurite outgrowth during development in vitro and in vivo [174,175,176].

In regeneration, macrophage-derived NOX2 complexes were shown to be incorporated into Rab7-positive endosomes of DRG neurons after injury, where they influence PI3K signaling to stimulate neurite outgrowth and axon regeneration after sciatic nerve lesion [177]. A genome-wide screen of factors limiting axon regeneration in mouse cortical neurons in vitro, identified that Rab27 negatively impacted axon growth and regeneration. Rab27 is involved in synaptic vesicle regulation and its silencing resulted in increased axon regeneration in *C. elegans* and increased axon regeneration after optic nerve crush and spinal cord injury in mice [178].

In summary, endosomes are key sites for the localization, recycling and targeting-for-degradation of growth-associated cargo. Regulation of endosome localization and transport in neurons can have key implications for axon growth and regeneration.

## 5. Lysosomes and Autophagosomes

### 5.1. Structure and Function

The lysosome is a dynamic organelle that supports multiple cellular functions. The classical view of lysosomes is that they function for the degradation and recycling of biological macromolecules that are delivered to the lysosome by autophagic, endocytic, and phagocytic routes. However, more recently, lysosomes and late endosomes have emerged as important compartments with roles as platforms for local protein synthesis, in the regulation of gene expression, cell growth, plasma membrane repair, synaptic plasticity, and other functions. Lysosomes are also involved in calcium signaling, lipid signaling, response to injury signaling, and importantly interaction with other organelles (reviewed in [179,180,181,182]). Furthermore, lysosomal/late endosomal trafficking is dynamically regulated, and the position of these organelles influences their function in neuronal compartments (reviewed in [180,183]). Autophagosomes participate in degradation through gathering cellular material and mitochondria then fusing with lysosomes, but are also an important part of the recycling machinery (reviewed in [184,185]).

### 5.2. Lysosomal Regulation of Autophagy in Axon Growth and Regeneration

The process of autophagy has been implicated in both neuronal survival and growth. The entry of molecules into autophagosomes can be non-selective, but the autophagic turnover of cell surface molecules involved in axon growth is selective, being regulated by ubiquitination, which is recognized by adaptors on autophagosomes (reviewed in [184,185]). In axons, autophagosomes have been observed to mature as they are retrogradely transported towards the cell body where they fuse with lysosomes [186,187,188]. The majority of autophagosomes are formed at the distal part of the axon near synapses [186,187,188,189]. Autophagosomes bud from two sources: Rab11-recycling endosomes [190] and the endoplasmic reticulum [188]. Importantly, the rate of autophagosome generation declines with neuronal maturation [191,192,193,194,195], coinciding with loss of regenerative ability. The fusion of autophagosomes with lysosomes was recently shown to play an important role in axon regeneration [196]. Failure of regeneration in inhibitory scar tissue is associated with the formation of dystrophic endbulbs on cut axons. These are induced by blockade of autophagic flux as autophagosomes fuse with lysosomes [196,197]. There are, however, contradicting reports on whether the manipulation of autophagy itself is beneficial or detrimental to the intrinsic growth potential of neurons. These studies are summarized in Table 1. The controversy is further complicated when considering that autophagy-related genes can influence growth via noncanonical pathways [198]. For instance, autophagy influences axon growth by the regulation of microtubule dynamics [199,200] whilst the autophagy-inducing kinases ULK1 and ULK2 have been confirmed to regulate axon growth by autophagy-independent mechanisms [201]. In addition, the function of autophagy differs among developmental maturation ages and in neuron types with different regeneration capacities ([189,199,202]; see also Table 1). Taken together, autophagy has been associated with axon growth and regeneration, functioning through the regulation of various mechanisms and molecules including adhesion complexes, the cytoskeleton, and general metabolism. Whether autophagosome function promotes or inhibits axonal regeneration after injury appears to depend on the status of the neuron, but further studies are needed to determine how autophagic mechanisms can be best targeted for regenerative gain.

### 5.3. Lysosome/Endosome Regulation of Intracellular Pathways in Axon Growth and Regeneration

Recent work has identified various ways in which lysosomes and late endosomes can affect axon growth and regeneration by acting as molecular platforms for key signaling pathways. One of the most well-studied pathways in this regard is the PI3K to mTOR pathway. mTOR (mammalian target of rapamycin) is a serine/threonine protein kinase that is localized on the outer surface of the lysosome. Acting through mTOR complexes 1 and 2, mTOR influences fundamental cell processes, leading to the regulation of protein synthesis, metabolism, and autophagosome generation (reviewed in [205,206,207]). There are many molecular mechanisms that influence the activation state of mTOR and are known to influence axon regeneration after injury in vivo. For instance, PTEN (phosphatase and tensin homolog), TSC1 (tuberous sclerosis complex 1), TSC2 (tuberous sclerosis complex 2) and GSK3 (glycogen synthase kinase) are negative regulators of mTOR activity and their genetic deletion in transgenic mice promoted axon regeneration in various models of traumatic nervous system injuries [18,19,208,209,210,211,212]. Axon regeneration in the CNS can also be achieved by delivering positive regulators of mTOR activity such as AKT [213], wnt10b (Wnt family member10b) [214], a specific mutant of HDAC (Class II histone deacetylase 5) [215], or PI3K delta [20] among others. Ageing is another important factor influencing mTOR. It was shown in RGCs and cortical neurons that mTOR signaling declines during maturation in line with their regeneration capacity [18,19,20]. Taken together, it is well established that mTOR activation in neurons is an important driver for axon regeneration. However, mTOR-activation and signaling occurs on the surface of late endosomes and lysosomes, so these organelles must be present in the tips of axons in order for local signaling to occur. In addition, mTOR was found to specifically localized at the growth cones of developing callosal projection neurons where it was necessary for axon extension [216].

### 5.4. Lysosome/Late Endosome Regulation of Exocytosis for Axon Growth and Regeneration

Exocytosis from lysosomes is another important mechanism that contributes to membrane repair and axon growth. As discussed already, intracellular free calcium is an important secondary messenger and one of its functions is to initiate exocytosis from lysosomes to other intracellular organelles and the extracellular space [217,218,219,220,221,222,223,224]. Exocytosis from lysosomes has been shown to contribute to neurite outgrowth in primary neuronal cultures derived from superior cervical ganglion [220], hippocampus [222], and cortex [224]. The Ca^2^-dependent exocytosis of lysosomes participates, together with ER, multivesicular bodies and other endosomes, in the addition of new membrane for axon growth and resealing of plasma membrane in case of an injury. Importantly, lysosomal exocytosis may also contribute to axon elongation by making the extracellular matrix more growth permissive by the secretion of lysosome-associated proteins. For instance, it was shown that lysosomal exocytosis results in the release of ATP [225] (which could act as an extracellular signaling molecule), the lysosomal enzyme acid sphingomyelinase [226] and various lysosomal cysteine proteases [223,227], which can modify the extracellular matrix.

In summary, autophagosomes and lysosomes/late endosomes contribute to axon growth and regeneration by the regulation of multiple processes including autophagy, intracellular signaling and exocytosis.

## 6. Proteasome

### 6.1. Structure and Function

The proteasome is a membrane-bound organelle which has an essential role in the clearance of excessive or damaged proteins [228]. It consists of several catalytic units which form a large cytoplasmic complex. Proteasomes recognize polyubiquitinated proteins and degrade them for clearance. Proteasomal distribution along the axon is key to proper neuronal development and maturation [229]. Mice lacking the adaptor protein PI31, which is essential for proteasome translocation to the distal axon, show disrupted synaptic structures and reduced survival highlighting the importance of the proteasome in protein degradation in axonal development [230]. In the axon, proteasomes are transported in anterograde and retrograde direction along microtubule tracts with the help of motor proteins with dynamics that resemble membrane vesicle movement and association with intracellular membranes [231]. At dendritic spines, fast activity-dependent recruitment of proteasomes has been implicated as an essential mechanism of protein degradation and this is important for the formation and maintenance of synapses (reviewed in [232,233,234]).

### 6.2. Protein Degradation by the Proteasome in Developmental Axon Growth

Proteasomes have an important role in neurite outgrowth. This was first described in the early 1990s, when application of the proteasome inhibitor lactacystin induced neurite outgrowth in PC12 cells and in neuroblastoma cell-lines [235,236,237]. Treatment of NGF in PC12s cells induces neurite outgrowth and was shown to reduce proteasomal activity by changing its catalytic unit composition. In primary neurons though, pre-treatment with proteasome inhibitors completely ablates axon formation and low concentrations of inhibitors reduced axon elongation and branching [238]. In addition, treatment of sympathetic and sensory neuronal explants with lactacystin, applied at the same concentrations as used for neurite outgrowth induction in PC12 cells, resulted in neurite extension block, which subsequently results in neurodegeneration [239]. These studies suggest that in neurons, basal level proteasome function is required for developmental axon growth. The role of the proteasome in neurodegeneration is extensive and is summarized elsewhere [240,241,242].

One mechanism through which proteasomal manipulation can influence neurite outgrowth is through the regulation of protein degradation at the growth cone. For example, Akt signaling phosphorylates radixin, a protein which tethers F-actin to the plasma membrane during growth, at a specific residue that protects it from proteasomal degradation. This resulted in stabilized interaction with F-actin and proper neurite outgrowth and growth cone formation [243]. In addition, the ubiquitin ligase Rnf6, which is expressed in both sensory and motor axons, specifically targets LIMK1 for degradation by the proteasome. LIMK1 plays an important role in regulation of the actin cytoskeleton during growth suggesting a role for the ubiquitin/proteasome system in regulating local cytoskeleton growth cone dynamics [244]. The proteasome also contributes to the establishment of neuronal polarity as inhibition of the proteasome results in a uniform rather than axon-confined distribution of Akt, which, in turn, results in the formation of multiple axons [245]. The ubiquitin-proteasome system was also recently implicated in axon guidance as Sema-3A, a secreted growth-repulsive guidance cue, was shown to promote FMRP (an RNA-binding protein involved in axon growth) proteasomal degradation in growth cones [246]. FMRP degradation in turn, led to growth cone collapse and turning away from repulsive cues [246].

### 6.3. Protein Degradation by the Proteasome in Axon Regeneration

The ability of axons to form a new growth cone and to regenerate after injury is dependent on local protein synthesis and degradation [74,247]. Application of protein synthesis and proteasome inhibitors to either whole or axon-only preparations of injured DRG or retinal axons in vitro prevented them from forming a new growth cone suggesting a role for protein turnover near the injury site for successful regeneration [74]. In ligated sciatic nerve, proteasomes accumulated proximally to the ligation but not distally, suggesting that the ligation obstructed anterograde axonal transport of the proteasome [231]. Interestingly, protein synthesis machinery was found at higher levels in PNS neurons compared to CNS neurons, whereas proteasome components showed the opposite distribution with higher levels in the CNS [74]. In fact, in conditioned DRGs which mount a robust regenerative response, the proteasome machinery load was reduced [74], indicating that protein balance is essential for axon regeneration. A similar decrease in axonal proteasome levels due to increased retrograde transport was observed early on in axon development of hippocampal neurons in vitro and in the cortex in vivo [248], further suggesting the localization of proteasomes in the axon could be inhibitory for growth. The removal of proteasomes from the newly growing axon can be triggered by pro-growth influences such as BDNF and cAMP suggesting the need for protein degradation downregulation in order to achieve protein stability during axon development [248]. Blockade of retrograde proteasomal transport resulted in proteasome accumulation at the axon tip and this reduced axonal growth [248]. Another growth factor—TGF-beta—was shown to inhibit proteasomal function, which increased neurite regeneration after scratch lesion in primary midbrain cultures [249]. Peripheral nerve injury was shown to evoke calcium influx into axons, which, in turn, activated the proteasome to degrade ubiquitinated proteins [250]. In particular, this activation of the proteasome at the site of injury resulted in neurofilament degradation and triggered Wallerian degeneration [250]. In a different injury paradigm, proteasomal activation due to transient axonal stretch injury primary cortical neurons prevented secondary injury, likely through effects on stabilizing the cytoskeleton. In this case, proteasomal activation was protective [251].

The studies above suggest that proteasomal activity in the axon requires a critical level of regulation and plays an essential role in protein balance in newly growing and regenerating axons. Further studies are needed to uncover how the proteasome can be targeted to promote growth and axon regeneration after injury.

## 7. Organellar Interconnections

Membrane-bound organelles allow the cell to perform multiple, incompatible biochemical processes in separated compartments, but at the same time require intricate levels of synchronicity and regulation. Inter-organellar interactions have been historically difficult to study due to their dynamic nature and due to their low temporal and spatial resolution. Only recently, with the advancement of high-resolution microscopy, could the complexity of this interconnectivity be studied [25,26]. In fibroblasts, multi-level interactions between six different organelles were recently observed by confocal and lattice light sheet microscopy [252]. These studies revealed that each organelle has a unique distribution pattern in the cell but at different points in time, some organelles come into contact with each other to perform a specific joint function [252]. Focused ion beam-scanning electron microscopy allowed for the 3D reconstruction of organelles in neurons and showed complex interactions between the continuous endoplasmic reticulum and an array of other organelles [31]. Membrane contact sites (MCS) are described as close membrane appositions (10–30 nm) between the outer lipid layers of two membrane-bound organelles and these are essential for inter-organellar signaling and function. MCSs have recently been investigated in the context of axon growth and regeneration [253,254]. MCS are not merely physical interactions between organellar membranes—they can often be enriched in various proteins and tightly regulated depending on the physiological conditions of the cell [255]. In fact, mutations in many proteins stabilizing MCSs have been described as causative in human axonal disease (Table 2) suggesting a role for organelle-organelle interactions in the processes of neuronal development, growth and regeneration.

### 7.1. ER-Mitochondria Interactions

The ER and mitochondria form contacts at mitochondrial-associated membranes (MAMs) and are essential for a number of cellular functions such as calcium homeostasis, regulation of apoptosis, lipid synthesis and trafficking and energy production (reviewed in [267,268,269,270,271,272,273,274]). Defective ER-mitochondria linkage has recently been implicated in several neurodegenerative conditions underlying its importance for axon maintenance [275,276,277,278,279]. Grp75 is a protein linker found at MAMs that initiates contact between IP3R in the ER and VDAC1 in the mitochondria. These contacts regulate calcium transfer between the two organelles [280]. Grp75 mRNA is upregulated upon axonal injury in the PNS [72] and local axonal synthesis of Grp75 is initiated as a response in primary hippocampal neurons after injury [253]. Grp75 deletion leads to abnormal axon development, while its overexpression prompts increased association between ER and mitochondria in injured hippocampal neurons in vitro and in sciatic nerves after sciatic nerve injury in vivo [253]. Both experiments showed that increased interconnection between ER and mitochondria improved calcium buffering and potentiated ATP production, which ultimately led to increased levels of axon regeneration and functional recovery after injury [253]. Interestingly, mitochondrial tethering to Grp75 at the growing tip of neurites improves axon outgrowth in cultured cortical neurons [281]. Another protein present at the ER-mitochondria interface is REEP1. REEP1, an ER-shaping protein responsible for axon growth and maintenance, but it also contains mitochondria-linking subdomains that may aid growth as well. Overexpression of disease-associated mutant REEP1 in mouse cortical neurons reduced dendritic tree complexity as a result of decreased ER-mitochondria tethering [45].

### 7.2. ER-Lysosomes/Late Endosomes Interactions

MCSs between endoplasmic reticulum and the endocytic pathways have emerged as regulators of endosomal function and localization, and their association is normally microtubule-associated and increases as endosomes mature [282,283]. Interestingly, ER tubule positioning along the axon and its morphology were recently shown to depend on motile lysosomes in response to chemical, light or metabolic stimuli [284]. Ablation of ER-lysosome MCS protein, VAP-A, resulted in ER fragmentation and reduced axonal length in growing RGC axons [284]. The molecular composition of MCSs between ER and lysosomes/late endosomes is still not very well studied. One protein shown to modulate the interaction between the ER and late endosomes (LE) is ER-resident protein Protrudin. Protrudin recognizes specific phosphoinositides and Rab7 on late endosomes and tethers them to the ER [285]. Protrudin then acts as an adaptor to facilitate the loading of kinesin-1 from the ER onto the motor adaptor—FYCO1—which is located on Rab7-positive LEs [285]. This process of repeated ER–LE contacts results in the anterograde transport of late endosomes along microtubules towards the cell periphery and is essential for protrusion formation and neurite outgrowth [285]. Overexpression of protrudin in RPE1 or PC12 cells induced long protrusions in a Rab7 and kinesin-dependent manner. Synaptotagmin-1 assisted this process by allowing the fusion of late endosomes and the plasma membrane. Over expression or activation of Protrudin in mature CNS axons also facilitates robust long-range regeneration, but whether this process relies on increased lysosomal transport has not been reported. The interaction between Protrudin and the ER MCS VAP proteins (VAP-A and-B) is required, because genetic interference with this interaction hinders the protrudin’s regenerative actions [43].

In addition, the endosomal protein IST1 and ER-localized isoform of spastin, a microtubule-severing protein, act at ER-endosome MCSs to promote endosomal fission [286,287]. Defects in this system lead to abnormal protein sorting and lysosomal dysfunction in primary cortical neurons and iPSC-derived neurons from HSP patients with spastin mutations [287]. As a result, the link between ER-regulated endosomal fission at MCSs and lysosomal function was proposed as a major mechanism of axon degeneration in HSP. In addition, ER-endosomal interaction has been shown to play an important role in receptor signaling. For example, once activated on the cell surface, the EGFR receptor is internalized in endosomes and continues signaling until dephosphorylated by PTP1B at ER-endosome contact sites [288,289]. ER-endosome contact sites were shown to be regulated by TPC1—an endo-lysosomal ion channel controlled by calcium signaling. Reduced MCS formation resulted in decreased endosome to ER contact and prolonged EGFR signaling, suggesting the role of MCSs as calcium-dependent hubs for signaling [290]. Receptor signaling timing and calcium influx play critical roles in axon growth and regeneration, but it is yet to be studied whether the mechanisms of ER-endosomes play a role in neurons. In addition, a novel regulator of ER-endosome contact sites was recently described. TMEM16K on the ER interacts with Rab7 on late endosomes to initiate MCS formation [291]. Loss of TMEM16K leads to impaired endo-lysosomal transport and function as well as neuromuscular and motor deficits in mice [291].

Recent studies implicated PDZD8, an ER-associated protein, and Protrudin together with the late endosomal protein Rab7 as main components of ER-endosome contact sites [292]. PDZD8 is also a component of the ER-mitochondria MCSs [293] and enables the recruitment of mitochondria to ER-endosomal MCS in order to regulate lipid transport and endosomal function [292]. Studying the significance of such complex inter-organellar interactions facilitated by multiple proteins in the context of axon growth and regeneration will be an interesting future area of research.

### 7.3. Endosome/Lysosome-Mitochondria Interactions

Endosome-mitochondria tethering also plays an important role in axon growth and integrity. One way through which this could occur is by regulation of protein translation. For example, RNA granules have previously been reported to tether to lysosomes which provide a platform for mRNA long-range transport along the axon [294]. In addition, Rab7-positive endosomes in contact with ribonucleoprotein particles often dock onto mitochondria in order to secure energy for local protein translation [30]. A mutation in Rab7, which is associated with Charcot-Marie-Tooth disease, results in defective binding of Rab7-endosomes to mitochondria, resulting in reduced protein synthesis in the axon and compromised axon integrity after initial axon outgrowth in RGCs [30]. Interestingly, ER is also found to be in close proximity to mRNA granules and late endosomes [30]. They form MCSs near processing bodies that carry mRNAs around the cells regulating their dynamics and translocation [295]. This all suggests a possible involvement of mitochondria and ER in endosomal trafficking or regulation of mRNA translation during growth which may have implications for regeneration in the CNS.

Mitochondrial interaction with the autophagosome-lysosome system is crucial for the process of mitophagy where depolarized mitochondria are actively degraded. Some studies point towards retrograde transport of damaged mitochondria to be coupled with mature lysosomes in the soma for degradation [296,297]. In fact, damaged mitochondria were shown to bulk release mitochondria-anchoring protein syntaphilin and these are transported to the cell body for degradation via interaction with late endosomes [298]. This process was found to be essential for survival in neurons upon ischemic injury and during neurodegeneration [298,299]. In addition, others have documented that dysfunctional mitochondria stalled in the distal axon, which suggests that local clearance might also occur independently of retrograde transport [300]. Indeed, in rat hippocampal neurons, triggering mitochondrial damage at physiological levels in the distal axon results in the recruitment of LC3-positive autophagosomes, LAMP1-positive lysosomes and Parkin, an E3 ubiquitin ligase for local degradation of damaged mitochondria [301]. As mentioned in Section 3, mitochondria depolarize after axonal injury, so their removal plays an essential role in axon regeneration. Further studies are needed to characterize the exact role of mitochondria-autophagosome-lysosome connection in axon growth and regeneration.

### 7.4. Proteasome-Other Organelles

A proportion of proteasomes in cultured neurons was also shown to form connections with lysosomes, which might play a role for the retrograde transport from dendritic spines to the cell body. In addition, the proteasome was shown to make contacts with synaptic vesicles, Golgi-derived vesicles and with mitochondria [231]. Recently, live-cell imaging in hippocampal neurons revealed that proteasomal inhibition impairs the axonal transport of APP by stimulating its trafficking to the endo-lysosomal system for cleavage. This study underlines the importance of proteasome-lysosome crosstalk and implicated proteasomal dysfunction in the abnormal APP metabolism causing axonal degeneration in some neurodegenerative diseases [302].

## 8. Conclusions

Axon growth and regeneration are cellular processes that involve signaling from the growth cone, or the site of injury back to the cell body via signaling endosomes. They, in turn, instruct changes in gene expression and axonal transport via the endosomal system as well as in protein synthesis, distribution and turnover through the ER, lysosome and autophagosome. This results in growth-related cargo translocation to the growth cone, the generation of energy supply through the mitochondria and intracellular signaling for membrane expansion and targeted growth. All these processes need to exist in concert to elicit successful growth and regeneration programs. Our understanding of cellular organelles has dramatically changed over the past decade from individual, isolated compartments, carrying out specific functions to highly interconnected and dynamic networks that regulate cascades of molecular events in three dimensions within the cell. Individually, organelles have been extensively studied in the context of axon growth and regeneration where they play key roles in the growth process by acting as molecular platforms to housing, distributing and regulating numerous intracellular molecules, materials and signaling components. Only recently with the advancement of high-resolution microscopy, the intricate interactions between different organelles or networks of organelles were made possible to study. We propose a model in which organelle contact sites orchestrate highly dynamic and interconnected series of molecular and cellular events that influence, and are needed, for axon growth and regeneration (Figure 1). Future efforts should be centered to studying these connections between organelles in the context of axon regeneration, where they could potentially serve as therapeutic targets to improve the intrinsic regenerative capacity of adult CNS neurons.

## Figures and Tables

**Figure 1 ijms-22-01798-f001:**
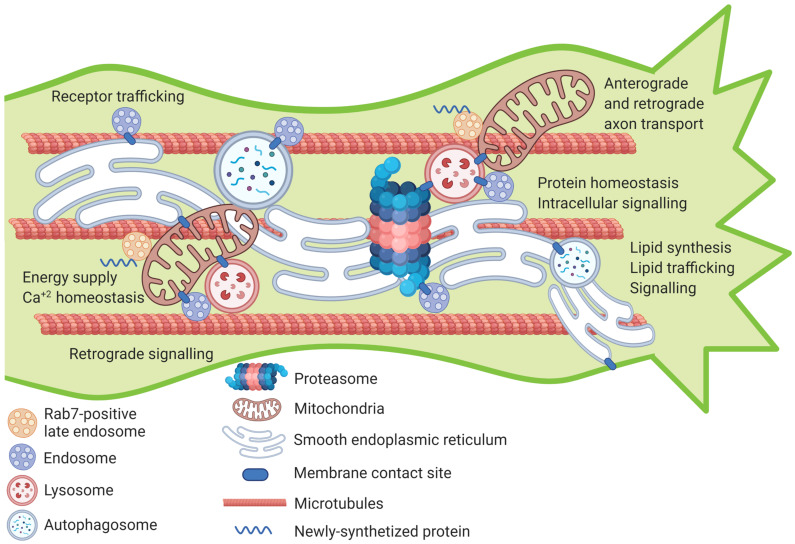
Proposed model for inter-organellar interactions within the axon shaft. Within the cell, various organelles form transient or permanent interactions with each other. Many of these organellar inter-connections have previously been documented in the axon (such as ER-mitochondria, ER-lysosome, ER-endosome, mitochondria-lysosome) but some are yet to be studied. This model proposes that membrane-contact sites between pairs of organelles or multiple organelles could occur in the axon shaft near the growth cone of developing or regenerating axons. These interactions play an essential role in numerous processes involved in axon growth and regeneration such as lipid signaling and trafficking, protein turnover, intracellular signaling, calcium buffering, receptor trafficking and energy homeostasis. Future studies on the developmental regulation of the quantity and dynamics of inter-organelle membrane contact sites could provide clues on how they can be targeted to boost axon growth and regeneration. Figure created with BioRender.com (www.biorender.com [last accesses 18 January 2021]).

**Table 1 ijms-22-01798-t001:** Summary of studies that assessed the effect of autophagy on neuronal growth.

Method for Manipulation of Autophagy	Neuronal Type Examined	Species and Age	Main Findings Regarding Neuronal Growth	Reference
Knockdown of ATG7 Application of 3-methyladenine	Primary cortical neurons	Embryonic rats and cultured for 1 till 3 days	Inhibition of autophagy resulted in: - elongation of neurites in vitro - reduction of RhoA signalling	Ban et al., 2013 [203]
Knockout of: ATG-2, ATG9, ATG13, EPG-8, IGG-1, UNC104	PVD nociceptive sensory neuron	Larval and adult *C. elegans*	Inhibition of autophagy resulted in: - elongation of the axon in vivo	Stavoe et al., 2016 [189]
Knockout of: WDR47	- Primary cortical and hippocampal neurons - Callosal and corticofugal neurons	- Embryonic mice and cultured for 4 days - Adult mice (16 weeks old)	Activation of autophagy resulted in: - Impaired formation and dynamics of growth cones in vitro -Defective and reduction of axonal projections in the corpus calossum in vivo - Destabilisation of microtubules	Kannan et al., 2017 [200]
Knockout of: ATG-2, ATG9, ATG13, EPG-8, IGG-1, UNC104	- HSN serotonergic motor neuron - DA9 cholinergic motor neuron - RIA interneuron - RIB interneuron - NSM pharyngeal neurosecretory-motor neuron	Larval and adult *C. elegans*	Inhibition of autophagy resulted in: - no phenotype in vivo	Stavoe et al., 2016 [189]
Application of Tat-beclin1	- Primary cortical neurons - Neuron types with axon fibres in the spinal cord	- Embryonic rats and cultured for 1 till 3 days - Adult mice (8 till 10 weeks old) subjected to spinal cord hemisection injury	Activation of autophagy resulted in: - enhanced neurite outgrowth on inhibitory substrates in vitro - Inhibition of axonal retraction after injury in cortical neurons in vitro and corticospinal neurons in vivo - Stimulation of axonal regeneration of monoaminergic neurons after injury in vivo - Stabilisation of microtubules	He et al., 2016 [199]
Application of 3-methyladenine	Dorsal root ganglion neurons	Adult rat (4 till 5 weeks old) and cultured for 1 day	Inhibition of autophagy resulted in: - Reduction of neuronal survival - Inhibition of neurite growth and branching	Clarke and Mearow, 2016 [204]

This table highlights that autophagosomes control axon growth and regeneration in neurons. The function of autophagy differs among neuron types and developmental maturation ages. However, the contradictory findings between the studies could also be the result of targeting different components of the autophagy machinery. It also cannot be excluded that the manipulated molecules influence axon growth via noncanonical pathways. The main results for each study are summarized and the colors indicate whether autophagy was considered negative (orange), had no effect (white), or positive (green) for axon growth and regeneration in the mentioned neuron type and age.

**Table 2 ijms-22-01798-t002:** Table summarizing human conditions caused by genetic deficits in genes important for MCSs between organelles.

Membrane Contact Site	Disease	Genes Involved	Gene Function	References
ER- Endosomes/ ER-PM/ER-Lysosome	Hereditary Spastic Paraplegia	Zfyve27	Vesicular membrane trafficking, ER-endosome/lysosome tethering	Mannan et al., 2006 [256]
ER-Endosome	Hereditary Spastic Paraplegia	Spastin	Microtubule-severing protein	Evans et al., 2006 [257]
ER-Mitochondria	Hereditary Spastic Paraplegia	REEP1	Microtubule-mitochondria	Zuchner et al., 2006; Beetz et al., 2008 [258,259]
ER-Endosome/ER-PM	Amyotrophic Lateral Sclerosis	VAP-A, VAP-B	ER-organelle tethering, facilitate protein interaction, vesicular trafficking	Nishimura et al., 2004 [260]
ER-Mitochondria	Early onset autosomal Parkinson’s disease	PARKIN, PINK1	Mitochondrial quality control and turnover	Kitada et al., 1998; Valente et al., 2004a; Valente et al., 2004b [261,262,263]
ER-Mitochondria	Charcot Marie Tooth Disease	MFN-2	Mitochondrial fusion, interaction with endoplasmic reticulum	Zuchner et al., 2004 [264]
ER-Mitochondria	Charcot Marie Tooth Disease	DNM-2	Vesicle trafficking, cytoskeleton dynamics, endosomal pathways	Sidiropoulos et al., 2012 [265]
ER-PM	Tubular Aggregate Myopathy	STIM-1	Calcium sensor	Bohm et al., 2017 [266]

This table highlights that mutations in genes which are important at MCSs between organelles could lead to human axonal disease. Additional gene functions are also listed.

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
