# Peer review of "Axonal Organelles as Molecular Platforms for Axon Growth and Regeneration after Injury"

_ijms, 2021, doi:10.3390/ijms22041798_

Round 1
Reviewer 1 Report
This review article by Petrova et. al., is detailed, timely, and contains most of the recent advancements in the axon regeneration field. It comes from the axon regeneration experts. Unlike any other axon regeneration review, this article specifically focuses on how axonal organelles act as platforms during axon development and axon regeneration process. This is an emerging field and I believe it will gain lot of attention from readers. The authors have done a very good job in covering most of the important discoveries related to axonal organelle and their roles in axon regeneration. Provided schematics is also informative. I strongly recommend publication of this article in IJMS.
Reviewer 2 Report
The review is dedicated to the molecular mechanisms that operate in axonal growth in embryogenesis and nerve regeneration after injury. The authors focuses on searching for new approaches to boost axon regeneration with a specific emphisis on spinal cord and optical injury.
The authors pay a special attention to organelles such as recycling endosomes, the endoplasmic reticulum, mitochondria, lysosomes, autophagosomes and the proteasome and their role in organizing the machinery for axon growth cone formation and axon regrowth in embryogenesis and nerve regeneration in vitro and in vivo.
Some sentences should be refrazed or corrected:
lines 201-201, 284-287, 243 and 261 contain repeated phrases, 440-445
